# Overview of Immune Checkpoint Inhibitors Therapy for Hepatocellular Carcinoma, and The ITA.LI.CA Cohort Derived Estimate of Amenability Rate to Immune Checkpoint Inhibitors in Clinical Practice

**DOI:** 10.3390/cancers11111689

**Published:** 2019-10-30

**Authors:** Edoardo G. Giannini, Andrea Aglitti, Mauro Borzio, Martina Gambato, Maria Guarino, Massimo Iavarone, Quirino Lai, Giovanni Battista Levi Sandri, Fabio Melandro, Filomena Morisco, Francesca Romana Ponziani, Maria Rendina, Francesco Paolo Russo, Rodolfo Sacco, Mauro Viganò, Alessandro Vitale, Franco Trevisani

**Affiliations:** 1Gastroenterology Unit, Department of Internal Medicine, Università di Genova, IRCCS (Istituto di Ricovero e Cura a Carattere Scientifico)-Ospedale Policlinico San Martino, 16132 Genoa, Italy; 2Department of Medicine and Surgery, Internal Medicine and Hepatology Unit, University of Salerno, 84084 Fisciano, Italy; 3Unità Operativa Complessa (UOC) Gastroenterologia ed Endoscopia Digestiva, ASST (Azienda Socio Sanitaria Territoriale) Melegnano Martesana, 20063 Milan, Italy; 4Multivisceral Transplant Unit, Department of Surgery, Oncology and Gastroenterology, Padua University Hospital, 35124 Padua, Italy; 5Gastroenterology Unit, Department of Clinical Medicine and Surgery, University of Naples Federico II, 80138 Naples, Italy; 6CRC “A. M. and A. Migliavacca” Center for Liver Disease, Division of Gastroenterology and Hepatology, Fondazione IRCCS Cà Granda Ospedale Maggiore Policlinico, Università degli Studi di Milano, 20122 Milan, Italy; 7Liver Transplantation Program, Sapienza University, 00185 Rome, Italy; 8Department of Surgery, Sant’Eugenio Hospital, 00144 Roma, Italy; 9Dipartimento Assistenziale Integrato di Chirurgia Generale, Unità Operativa Complessa Epatica e Trapianto Fegato, Azienda Ospedaliera Universitaria Pisana, 56126 Pisa, Italy; 10Internal Medicine, Gastroenterology and Hepatology, Fondazione Policlinico Universitario A. Gemelli IRCCS, 00168 Rome, Italy; 11UOC Gastroenterologia Universitaria, Dipartimento Emergenza e trapianti di organo, Azienda Policlinico-Universita’ di Bari, 70124 Bari, Italy; 12Gastroenterology and Multivisceral Transplant Unit, Padua University Hospital, 35124 Padua, Italy; 13UOC Gastroenterologia ed Endoscopia Digestiva, Azienda Ospedaliera Universitaria “Ospedali Riuniti”, 71122 Foggia, Italy; 14Division of Hepatology, Ospedale San Giuseppe, University of Milan, 20122 Milan, Italy; 15UOC di Chirurgia Epatobiliare, Dipartimento di Scienze Chirurgiche Oncologiche e Gastroenterologiche, Azienda Università di Padova, 35124 Padua, Italy; 16Dipartimento di Scienze Mediche e Chirurgiche Alma Mater Studiorum, Università di Bologna, 40126 Bologna, Italy; 17HCC Special Interest Group, Associazione Italiana per lo Studio del Fegato (AISF), 00199 Roma, Italy

**Keywords:** check-point inhibitors, liver disease, immunotherapy, outcome

## Abstract

Despite progress in our understanding of the biology of hepatocellular carcinoma (HCC), this tumour remains difficult-to-cure for several reasons, starting from the particular disease environment where it arises—advanced chronic liver disease—to its heterogeneous clinical and biological behaviour. The advent, and good results, of immunotherapy for cancer called for the evaluation of its potential application also in HCC, where there is evidence of intra-hepatic immune response activation. Several studies advanced our knowledge of immune checkpoints expression in HCC, thus suggesting that immune checkpoint blockade may have a strong rationale even in the treatment of HCC. According to this background, initial studies with tremelimumab, a cytotoxic T-lymphocyte-associated protein 4 (CTLA-4) inhibitor, and nivolumab, a programmed cell death protein 1 (PD-1) antibody, showed promising results, and further studies exploring the effects of other immune checkpoint inhibitors, alone or with other drugs, are currently underway. However, we are still far from the identification of the correct setting, and sequence, where these drugs might be used in clinical practice, and their actual applicability in real-life is unknown. This review focuses on HCC immunobiology and on the potential of immune checkpoint blockade therapy for this tumour, with a critical evaluation of the available trials on immune checkpoint blocking antibodies treatment for HCC. Moreover, it assesses the potential applicability of immune checkpoint inhibitors in the real-life setting, by analysing a large, multicentre cohort of Italian patients with HCC.

## 1. Introduction

Hepatocellular carcinoma (HCC) is the sixth most common cause of cancer, and ranks fourth among the causes of cancer-related death [1]. Major risk factors for HCC include chronic infection with the hepatitis C (HCV) and B (HBV) viruses, heavy alcohol drinking, and aflatoxins B1 exposure, depending on geographical epidemiology. In recent years, Non-Alcoholic Fatty Liver Disease (NAFLD), the hepatological aspect of the metabolic syndrome, has been recognised as a relevant cause of advanced chronic liver disease, and the fastest growing cause of cirrhosis and HCC in Western countries [2]. Although mixed data exist about the exact magnitude of HCC risk in patients with NAFLD, and different epidemiological and methodological confounders must be taken into account, in a recent retrospective cohort study involving 130 facilities in the United States Veterans Administration health service, Kanwal et al. found that the risk of HCC was higher in NAFLD patients than in the general population, with a 5- and 10-year cumulative incidence rate of HCC of 0.8 and 1.7 per 1000 patients in NAFLD patients as compared to 0.09 and 0.18 per 1000 patients in controls [3].

HCC represents a unique and peculiar neoplastic setting, as in up to 80% of cases it arises on the background of cirrhosis and chronic inflammation, which is now considered an important factor involved in cancer progression [4]. Indeed, liver cirrhosis is a recognised model of local chronic inflammation driven by infiltrating immune cells and resident liver cells like Kupffer cells, dendritic cells, liver sinusoidal cells and hepatic stellate cells. Chronic inflammation initiates tissue remodelling and determines an oxidative microenvironment, triggering DNA damage and genomic aberrations that eventually culminate in neoplasia, and as a fact it is recognised that cirrhosis and chronic inflammation act as a favourable preneoplastic setting [4]. Although precise molecular links between inflammation and HCC have not yet been fully elucidated, most data rely on the activation of the tumour necrosis factor-nuclear factor-κB axis, transcription target STAT3 and janus kinases activation as procarcinogenetic in the liver, while another player recently identified in this field is the inflammasome, a multiprotein complex and sensor of cellular damage [5,6]. Thus, as in no other neoplasia, the development of HCC is a multi-event process involving a series of genetic mutations (pr3RB, β-catenin, chromatin and transcription modulation) and epigenetic events such as hystone acetylation/deacetylation leading to a dysregulation of various genes, which may also represent putative therapeutic targets [7,8].

In the past fifteen years, advances in molecular and tumour biology significantly modified the paradigm of cancer treatment, moving from a histopathological basis to targeting specific molecular patterns. This review focuses on HCC immunobiology and the rationale for immune checkpoint blockade in these patients, while a specific discussion has been dedicated to a critical evaluation of the available trials on immune checkpoint inhibitors, alone or with other therapies, for HCC. Lastly, we assessed the potential applicability of immune checkpoint inhibitors to the real-life setting analysing a large cohort of Italian patients with HCC.

## 2. Cancer Immunotherapy

The principle of tumour immune surveillance presumes that most pre-malignant and early malignant cells can be eliminated (or controlled) by the immune system [9]. However, a critical feature of advanced tumours compared to early malignant lesions is their ability to escape adaptive immune response. During malignant transformation, tumour-associated antigens generated by gene mutations are created and recognised by the immune system, and adaptive tumour antigen-specific T-cell responses are generated, leading to cancer-cell elimination [10]. Therefore, to survive, growing tumours must adapt to their immunological environment by either turning off immune responses, and/or creating a local microenvironment that inhibits immune cell tumouricidal activity.

In normal circumstances, T-cells with a different T-cell receptor (TCR) repertoire circulate in the body patrolling for evidence of foreign peptides presented on the surface of cells due to infection or cancer development. The identification of tumoural antigen by T-cell determines an activation, with clonal proliferation/expansion, and a cytolytic response. On the other hand, the immune system plays a critical role in promoting tumour progression. This dual role by which the immune system can suppress and/or promote cancer growth is termed “cancer immunoediting” and consists of three phases: elimination, equilibrium, and escape [11].

In cancer immunotherapy, agents such as interferon, interleukins, vaccines and oncolytic viruses are used to enhance the immune system activation to attack tumoural cells through natural mechanisms. In particular, this goal can be achieved with several drug classes: checkpoint inhibitors, lymphocyte-promoting cytokines, engineered T-cells such as Chimeric Antigen Receptor T-cell (CAR-T) and TCR T-cells, agonistic antibodies against co-stimulatory receptors, and cancer vaccines [12]. The efficacy of cancer immunotherapies has been demonstrated, determining the rapid integration of these treatments into clinical practice. Moreover, one of the most attractive features of many cancer immunotherapies is that they target malignant cells and spare normal tissues from the damage often seen with radiation and chemotherapy that contributes to patient morbidity and mortality [13]. These properties of immunotherapy have supported the rapid inclusion of such a treatment into clinical practice. Currently, antibodies targeting cytotoxic T-lymphocyte-associated protein 4 (CTLA-4) (tremelimumab and ipilimumab) and the programmed cell death protein 1 (PD-1) or its ligand PD-L1 (nivolumab, pembrolizumab, atezolizumab, and durvalumab) have been approved for different types of solid tumours and acute lymphoblastic leukaemia.

However, despite continue advances in the field of cancer immunotherapy, several problems remain unsolved, including the inability to predict treatment efficacy, the need for additional biomarkers able to guide treatment, the development of cancer resistance immunotherapies, the lack of clinical study designs optimised to determine efficacy, and the high cost of treatment [14]. Moreover, due to the limited results in terms of efficacy and the narrow therapeutic index of some of these drugs, the adoption of a personalised pharmacogenetic approach would represent a turning point to improve results [15]. Even though all these findings are particularly relevant in HCC tissue, the limited efficacy of systemic therapies in HCC patients, and their poor tolerability to anticancer drugs, prompted the exploration of the potential of immunotherapy even in this setting, where immunotherapy is expected to play a pivotal role in the near future.

## 3. Rationale of Immune Checkpoint Blockade in Hepatocellular Carcinoma

In the last decade, many basic science advancements and discoveries related to tumour biology have been achieved by transcriptomic, genomic and epigenomic studies [16,17]. However, in the case of HCC, they poorly translated into clinical practice and only suboptimal results have been obtained in clinical trials testing many drugs in the last decade. As a result, although targeted systemic therapies for HCC provided some clinical benefits, the improvement in patient outcome remains modest, and HCC remains a difficult-to-cure tumour for various reasons: firstly, 70–80% of cases occur in the context of liver cirrhosis; secondly, intra-tumour morphologic and genetic heterogeneity make difficult our understanding of liver cancer, and may determine the resistance to targeted therapies; and, thirdly, either drivers or passengers mutations can be present in the tumour, making an effective molecularly-targeted therapy quite difficult [16,17,18]. This can explain why, despite good rationale and promising Phase II data, drug development in Phase III trials failed in many instances.

Currently, the standard-of-care for first-line treatment for advanced HCC is represented by two multikinase inhibitors (sorafenib and lenvatinib), and in patients who fail first-line treatment with these drugs, the second-line treatment is again represented by multikinase inhibitors (regorafenib and cabozantinib) [19,20,21,22]. The survival benefit obtained with multikinase inhibitors over the best supportive care is limited, and their tolerability is generally poor, indicating the urgent need for more efficacious and better tolerated therapeutic approaches.

One of the alternative strategies against the tumour relies on the modulation of the already existing immune response through the enhancement of activators and the block of inhibitors. T-cell exhaustion, defined as an impaired T-cell capacity to secrete cytokines and proliferate, with overexpression of immune checkpoint receptors (e.g., PD-1, CTLA4, and lymphocyte-activating 3) has been observed in certain types of cancer, including HCC [23]. Immune inhibitory receptors and ligands play a major role in induction and maintenance of HCC immune tolerance [24,25,26]. In particular, CTLA-4 is essential for the activation of helper CD4+ T-cells and the priming phase of the immune response. Upon binding of its ligands, CTLA-4 decreases T-cell activation following antigen presentation. CTLA-4 also plays a major role in the function of regulatory T-cells (Treg), a subset of CD4+ T-cells that inhibit the immune response. Moreover, CTLA-4 expression on CD14+ dendritic cells was associated with IL-10 and indoleamine-2,3-dioxygenase (IDO)-mediated inhibition of T-cell proliferation and induction of T-cell apoptosis [26]. In HCC patients, high CTLA-4 expression on Tregs in peripheral blood has been reported in association with a decrease in cytolytic granzyme B production by CD8+ T-cells [27]. Another immune checkpoint pathway is the one regulated by PD-1 receptor. PD-1 is a key factor in the effector phase of the immune-response, and is expressed by activated T- and B-cells and other cell types such as in the skin and in the lung: upon binding to its ligands (PD-L1 and PD-L2), PD-1 inhibits T-cell activation and proliferation [28]. The increased expression of PD-1 has been reported on CD8+ T-cells in patients with HCC, as well as an increase in tumour infiltrating and circulating PD-1+CD8+ T-cells associated with disease progression after curative hepatic resection [24,29]. In addition to the upregulation of PD-1 on T-cells, its ligand PD-L1 is highly expressed on peritumoural stroma cells as well as cancer cells, promoting a PD-L1/PD-1 pathway-driven inhibition of anti-tumour T-cell responses [29,30,31].

From a clinical standpoint, there is evidence of a role played by an activated immune-response in HCC behaviour: (i) the infiltration of T-cells in the tumour is correlated to neoplastic recurrence after liver transplantation; (ii) the presence of different immune cells infiltrating the tumour have been correlated to patients’ survival; and (iii) different immune-subtypes of the tumour microenvironment are variously associated with histological and molecular classification of HCC—with potential prognostic implications—and the presence of exhausted T-cells was associated with poorer patient survival [32,33,34,35,36].

The longer experience accrued with immunotherapy for other tumours is essential to guide clinicians in the HCC landscape. It is known that expression of tumour-infiltrating lymphocytes, features of inflammatory cells (PD-1 and PD-L1 expression), percentage of mutations in tumour cells and gene expression profiles correlate with the activity and efficacy of these drugs against several tumour types [37,38,39,40]. In selected neoplasms, tumour mutational burden measured by targeted next-generation sequencing panels or by whole-exome sequencing, may predict clinical response to immunotherapy [41,42]. Tumours with high rate mutations present highly immunogenic antigens and more immune infiltration and they are more suitable to be managed with immune checkpoint inhibitors. Conversely, tumours with lower mutational burden present less immunogenic antigens and lower immune infiltration and, therefore, they are better candidates to other therapies [43,44]. Recently, Samstein et al. analysed the genomic data (targeted next generation sequencing) of patients with several tumour types (but not HCC) treated with immunotherapy or other therapies. Among all patients, higher somatic tumour mutational burden (highest 20% in each histology) was associated with better overall survival, but the tumour mutational burden cut-points associated with improved survival varied markedly among tumour types, indicating that there may not be one universal definition of high tumour mutational burden [45]. Moreover, Sia et al. focused their attention on HCC and its microenvironment (interactions among tumour cells, immune cells, and other immunomodulators present in the microenvironment) showing that 25% of HCC have markers of an inflammatory response, with high expression levels of PD-L1, markers of cytolytic activity, and fewer chromosomal aberrations [46]. The authors called this group of tumours the “immune class”, and subdivided this class in two subtypes, characterised by active or exhausted immune response, the latter representing the ideal one to receive immunotherapy. Conversely, Harding et al. reported that HCC “cold” tumours (with Wnt/CTNNB1 mutations) are refractory to immune checkpoint inhibitors [47].

Better characterisation and understanding of increased immune checkpoints expression provide the rationale for the use of immune checkpoint blocking antibodies in HCC treatment. Figure 1 reports a schematic representation of the potential factors involved in immune system paralysis in HCC patients, and the potential pathways of action of various drugs. Binding the targeted molecules, the immune checkpoint inhibitors block the signalling, putting the immune response on hold, and allowing cytotoxic T-cells to strike tumour cells. Many Phase III trials testing the efficacy of monoclonal antibodies that target this pathway in HCC patients are ongoing, but the encouraging results reported in Phase I investigations has spurred the approval by FDA of immunotherapy even for this cancer [48,49]. Indeed, this therapy is the most interesting approach proposed according to the new discoveries in HCC biology, and especially the knowledge that liver has developed intrinsic tolerogenic mechanisms within the innate and adaptive immune system as a result of its constant exposure to antigens from portal-venous blood [50]. To date, all immune checkpoint targeted therapies for HCC consist of monoclonal antibodies developed for a specific immune target. Although several immune checkpoint blocking agents were identified in preclinical models, the majority of clinically tested therapies rely on antibodies targeting PD-1, PD-L1 and CTLA-4 molecules. The first small Phase II clinical trial using an immune checkpoint inhibitor, tremelimumab (a CTLA-4 blocking monoclonal antibody), targeted patients with HCC and chronic HCV infection, including a significant proportion (42.9%) of patients in Child-Pugh stage B [51]. A notable disease control rate (76.4%) was observed and the safety profile was acceptable. In a second small pilot trial, tremelimumab was combined with (incomplete) tumour ablation using locoregional therapies with the aim to synergise the effects by inducing immunogenic tumour cell death [52]. In this study, all aetiologies patients were included, liver function was preserved in the most patients, and 26.3% achieved a confirmed partial response. This study represented a proof of concept that immunotherapy in combination with tumour ablation is a potential way to treat patients with advanced HCC, and leads to the accumulation of intratumoural CD8+ T-cells. In fact, in tumour biopsies performed at six weeks, a clear increase in CD8+ T-cells occurred in patients showing a clinical response.

In patients with advanced HCC, PD-1 antibodies (nivolumab and pembrolizumab) have shown promising efficacy in therapy-naïve, as well as pre-treated patients. However, only 10–20% of them showed an objective and durable response. Therefore, combination schedules including different immune-therapies, (e.g., PD-1/PD-L1 and CTLA-4 antibodies) or the combination of immunotherapy and small molecules, or bifunctional antibodies are likely needed to improve response rates.

## 4. Strategies for Patients Selection

### 4.1. T-Cell Exhaustion

To select patients who are likely to clinically benefit from immune checkpoint inhibitors and to establish optimal strategies, a better understanding of T-cell exhaustion in the HCC microenvironment is crucial. It is known that pro-inflammatory cytokines such as IL-1β, IL-6, IL-8, IL-12, IL-18, and IFN-γ have been shown to enhance T-cell response, while anti-inflammatory ones, e.g., TGF-β and IL-10, promote T-cell exhaustion and infiltration in tumours [53].

The exhaustion profile of tumour-infiltrating CD8+ T-cells in HCC patients needs to be characterised in detail regarding heterogeneous subsets of exhausted T-cells. A recent study suggested that combination blockade of immune checkpoint receptors additionally restores the functions of tumour-infiltrating T-cells from HCC patients, although the identification of HCC patients eligible for a combined approach remains unclear [23]. Interestingly, Kim et al. investigated the heterogeneity of exhausted tumour-infiltrating CD8+ T-cells and the relationship with clinical features of HCC, focussing on the different molecular and cellular characteristics of the tumour-infiltrating CD8+ T-cell subpopulations, distinguished by differential PD-1 expression. They demonstrated that HCCs with a discrete population of PD-1-high CD8+ T-cells might be more susceptible to combined immune checkpoint blockade–based therapies [54]. Recently, Feun et al. performed a correlative study to investigate the correlation between circulating biomarkers and response to pembrolizumab. They found that the mean plasma TGF-β levels in responders were lower than in non-responders, and that a TGF-β level ≥ 200 pg/mL was an indicator of poor response to treatment. Furthermore, low baseline plasma levels of TGF-β were significantly associated with improved overall survival and progression free survival after treatment with pembrolizumab. These results support a study showing that TGF-β signalling diminishes tumour response to PD-1/PD-L1 blockade by excluding CD8-positive effector T cells from the tumour parenchyma [55,56].

### 4.2. The Gut Microbiota

The gut microbiota is a well-known modulator of the immune response and is able to mediate the response to immunological treatments, as shown in patients with melanoma, renal tumour and non-small cell lung cancer [57,58,59]. Recent studies also provided evidence that the gut microbiota is linked with the pathogenesis of HCC. In animal models of HCC, the correlation between circulating levels of inflammatory mediators and lipopolysaccharides (LPS) and the number and size of tumours suggests an interplay between the outgrowth of harmful bacteria, such as Gram-negative ones, and tumourigenesis [60,61,62]. Administration of antibiotics and probiotics or blocking the expression of toll-like receptor-4 (the LPS receptor) not only inhibits tumour cells proliferation but also reduces the infiltration of macrophages and the expression of tumour necrosis factor (TNF)-alpha and IL-6 in the liver tissue (Figure 1) [62,63].

In cirrhotic patients with HCC and non-alcoholic fatty liver disease (NAFLD), an altered gut microbiota profile, consisting in the reduction of beneficial and anti-inflammatory bacteria such as *Akkermansia* and *Bifidobacterium* and the increase of harmful ones such as *Enterobacteriaceae* and *Ruminococcus*, was associated with a pronounced intestinal inflammation that, in association with the increased intestinal permeability typical of cirrhotic patients, led to a systemic inflammatory response [64]. In these patients, an increase in circulating activated monocytes and monocytic myeloid-derived suppressor cells expressing PD-1 and PD-L1 points out that the persistence of an inflammatory stimulation derived from the gut eventually results in the paralysis of the immune system, favouring the process of hepatocarcinogenesis [65].

Based on these data, it is conceivable that the gut microbiota is implicated in the pathogenesis of HCC through immunostimulating and immunosuppressive mechanisms. Consequently, it can be expected that the response to immunotherapy might be modulated by the microbiota composition of HCC patients. The identification of a microbial signature associated with the response to immunotherapy could allow implementing modulation strategies, such as faecal microbial transplantation or the use of prebiotics, probiotics or postbiotics to personalise the therapeutic approach and maximise its effectiveness. This is an exciting and important field of future research aimed at improving the results of immunotherapy in HCC patients.

## 5. Outcome of Current Studies on Immunotherapy in Patients with HCC

### 5.1. Efficacy

Checkpoint inhibitor-based treatments will be, in the near future, an important enrichment of the therapeutic armamentarium against HCC, and probably not only as first/second line approach to advanced stage tumours as a single or combined systemic therapy, but also in early and intermediate stages in combination with surgery and locoregional treatments. The first drugs of this class tested in HCC were tremelimumab, a CTLA-4 inhibitor, and nivolumab, a PD-1 antibody [48,51]. Until now, various other drugs have been tested: CTLA-4 antibodies ipilimumab, pembrolizumab, spartalizumab, tislelizumab and camrelizumab with a strong PD-1 inhibitory activity, and PD-L1 antibodies durvalumab, avelumab and atezolizumab [65]. The ongoing clinical trials exploring immune checkpoint inhibitors alone, or in combination with other drugs or with local therapies are summarised in Table 1.

### 5.2. Results of Monotherapy with Checkpoint Inhibitors in HCC

#### 5.2.1. Tremelimumab

In a Phase II open-label, multicentre clinical trial, Sangro et al. treated with this drug, at a dose of 15 mg/kg IV every 90 days until tumour progression or severe toxicity, 21 patients with HCV-related HCC (57% with an advanced stage and 76% naïve to sorafenib) [51]. Objective response and disease control rate were 76.4% and 17.6%, respectively. Median time-to-progression (TTP) was 6.48 months (95% CI: 3.95–9.14). No toxicities requiring systemic steroid treatment were recorded. These initial results on safety profile and antitumour activity in patients with advanced HCC supported subsequent studies.

#### 5.2.2. Nivolumab

A Phase I/II trial open-label, non-comparative, dose escalation and expansion trial (CheckMate 040) for the anti-PD-1 antibody nivolumab against HCC was completed [47]. In this trial, patients naïve to sorafenib, sorafenib intolerant or sorafenib refractory were treated with nivolumab at dose of 0.1–10 mg/kg once every two weeks (dose-escalating cohort) or at a dose of 3 mg/kg once every two weeks (expansion cohort). In a total of 262 patients, nivolumab 3 mg/kg showed, in the dose-expansion phase and in the dose-escalation phase, a manageable safety profile and an objective response rate of 20% (95% CI 15–26) vs. 15% (95% CI 6–28), and an overall survival at nine months of 74% vs. 66%. Despite these favourable data, preliminary results of a Phase III trial of nivolumab vs. sorafenib in first-line treatment (NCT02576509, CheckMate-459) showed that the study did not meet its primary end-point of overall survival [Hazard Ratio = 0.85 (95% CI: 0.72–1.02); *p* = 0.0752] [66,67].

#### 5.2.3. Pembrolizumab

This anti-PD-1 antibody is being developed primarily as a second-line treatment. In a non-randomised, multicentre, open-label Phase II trial (KEYNOTE-224, NCT02702414), pembrolizumab (200 mg intravenously every three weeks for about two years or until disease progression, unacceptable toxicity, patient withdrawal, or investigator decision), was administered, at three-week intervals, to 104 Child–Pugh class A sorafenib-refractory or sorafenib-intolerant patients. The interim results of this trial showed an objective response in 18 patients (17%; 95% CI 11–26) and a median survival of 12.9 months. The best overall responses were one (1%) complete and 17 (16%) partial responses; meanwhile, 46 patients (44%) had stable disease, 34 (33%) had progressive disease, and 6 patients (6%) who did not have a post-baseline assessment were considered not to be assessable. Treatment-related adverse events occurred in 76 patients (73%), which were serious in 16 (15%). Immune-mediated hepatitis occurred in three (3%) patients, but there were no reported cases of viral flares. According to the trial, pembrolizumab was effective and tolerable in patients with advanced HCC who had previously been treated with sorafenib and that the drug might be a treatment option for these patients [68].

In the global Phase III trial allocating patients with advanced HCC who were previously treated with systemic therapy to pembrolizumab or best supportive care (KEYNOTE-240, NCT02702401), pembrolizumab improved overall survival (Hazard Ratio: 0.78; one sided *p* = 0.0238) and progression-free survival (Hazard Ratio: 0.78; one sided *p* = 0.0209), although these differences did not meet significance per the prespecified statistical plan [69]. In the second ongoing, double-blind, randomised Phase III trial (KEYNOTE-394, NCT03062358), pembrolizumab is being tested against placebo in Asian patients with advanced HCC who previously received systemic therapy, having as primary endpoints progression-free and overall survival.

#### 5.2.4. Camrelizumab

A Phase II/III trials is ongoing with this anti-PD-1 antibody in China, enrolling patients with failure or intolerance to prior systemic treatment. Two-hundred seventeen patients were randomised (1:1) to camrelizumab 3 mg/kg iv for q2w (*n* = 109) or q3w (*n* = 108). Interim results showed an objective response rate of 13.8% (95% CI 9.5–19.1) (30/217) and six-month overall survival rate of 74.7%. Median time to response was two months (range: 1.7–6.2). Of the 30 responses, 22 were ongoing, and median duration of response was not reached. Disease control rate was 44.7% (95% CI 38.0–51.6), median time to progression was 2.6 months (95% CI 2.0–3.3), and median progression-free survival was 2.1 months (95% CI 2.0–3.2). The most common treatment-related adverse events were reactive cutaneous capillary endothelial proliferation (66.8%, all grade ≤ 2), increased aspartate (24.4%) or alanine aminotransferase (23.0%), and proteinuria (23.0%). Camrelizumab showed high objective response rate, durable response and acceptable toxicities in Chinese pretreated advanced HCC patients [70].

#### 5.2.5. Tislelizumab

A Phase I trial recruiting including 61 patients with various solid cancers (including HCC) showed safety profile (NCT02407990). In a Phase III trial started in December 2017, patients with HCC were allocated to tislelizumab (200 mg iv for q3w) or sorafenib (400 mg bid) as first-line treatment. The primary endpoint is overall survival and this trial is designed to consider the non-inferiority of tislelizumab compared to sorafenib. The study opened to accrual in December 2017 and is currently recruiting patients; approximately 640 patients will be recruited from approximately 100 sites globally [71].

#### 5.2.6. Durvalumab

A Phase I/II trial of durvalumab monotherapy for solid cancers, including a cohort of 30 patients with HCC, was completed (NCT01693562). A 10% objective response rate and a median survival time of 13.2 months were observed [72].

### 5.3. Combination of Two Immune Checkpoint Inhibitors

As previously reported, the anti-PD-1/PD-L1 and anti-CTLA-4 antibodies are expected to be promising agents in HCC immunotherapy not only as single agents, but also by combined with agents that have different targets. Therefore, several clinical trials evaluating the simultaneous blockade of multiple immune checkpoints are currently ongoing (Table 1). The high efficacy of combination therapy has already been shown in other solid tumours. For instance, the inhibition of the PD-1/PD-L1 pathway alone might not activate tumour immunity as expected if the required CD8+ T cells are not adequately represented in the tumour microenvironment. However, simultaneous inhibition of the B7-CTLA-4 pathway by an anti-CTLA-4 antibody may increase the number of activated CD8+ T cells in lymph nodes, followed by an increase in the number of activated CD8+ T cells infiltrating into tumoural tissues, thereby enhancing their antitumour effects. In addition, anti-CTLA-4 antibody therapy may be effective against regulatory T cells in the cancer immunosuppressive microenvironment.

#### 5.3.1. Durvalumab plus Tremelimumab

This combination, tested in a Phase I/II study in 40 patients, reported a 15% objective response rate, demonstrating that combined therapy is more effective than durvalamab alone [73]. This combination also showed manageable safety profile. Currently, a Phase III is ongoing to compare different regimens as a first-line treatment; the four arms consist of durvalumab monotherapy, two types of durvalumab plus tremelimumab combination therapies (regimens 1 and 2) and sorafenib monotherapy (NCT03298451) [74].

#### 5.3.2. Nivolumab plus Ipilimumab

A sub-cohort of the CHECKMATE-040 study is evaluating the combination of nivolumab plus ipilimumab in sorafenib-treated patients (NCT01658878). Preliminary results showed an objective response rate of 31%, with a median duration of response of 17 months and a median overall survival that varies between 12 and 23 months according to the different treatment schedules applied in the three arms of the study [75]. There are two other Phase II clinical studies evaluating this combination regimen: one of these studies is comparing, in USA, nivolumab monotherapy with nivolumab plus ipilimumab (NCT03222076), while the second study is evaluating, in Taiwan, the combination therapy alone (NCT03510871).

### 5.4. Combination of Immune Checkpoint Inhibitors with Molecular-Targeted Agents

The therapeutic outcomes of immune checkpoint inhibitors with molecular target therapies has demonstrated to be superior to those of monotherapy in other solid tumours. Therefore, even for HCC, therapies involving an immune checkpoint inhibitor plus a molecular targeted agent was suggested as a promising strategy in recent years. In particular, interstitial cells (Kupffer cells, dendritic cells, liver endothelial cells, and liver stellate cells) and immunosuppressive cytokines (e.g., IL-10 or TGF-β) may contribute to the immunosuppressive environment of HCC, and the PD-1/PD-L1 pathway plays an important role in the development of the immunosuppressive microenvironment in HCC. Combining a molecular targeted agent and an immune checkpoint inhibitor is expected to improve this immunosuppressive microenvironment.

#### 5.4.1. Atezolizumab plus Bevacizumab

A Phase III randomised controlled trial of atezolizumab plus bevacizumab versus sorafenib as a first-line treatment was started and is currently ongoing to confirm the results of the Phase Ib trial [76]. Preliminary results of this Phase III study (NCT03434379) have recently been released, and the combination of atezolizumab (1200 mg on day 1 of each 21-day cycle, intravenously) plus bevacizumab (15 mg/kg on day 1 of each 21-day cycle, intravenously) met both co-primary end-points of improvement in overall and progression-free survivals as compared with sorafenib (400 mg twice per day, on days 1–21 of each 21-day cycle), although survival figures have not yet been communicated [77].

#### 5.4.2. Pembrolizumab plus Lenvatinib

A Phase I trial for this therapy is also underway in patients with HCC. According to preliminary results, 46% of patients had either partial response or stable disease in the mRECIST criteria among the patients who had been evaluated [78].

#### 5.4.3. Other Combinations

Currently there are several early stage clinical studies considering various combination of PD-1 inhibitors and targeted agents for HCC, without available data for the moment. They include: nivolumab plus lenvatinib (NCT03418922), nivolumab plus cabozantinib (NCT03299946), nivolumab plus bevacizumab (NCT03382886), pembrolizumab plus regorafenib (NCT03347292), pembrolizumab plus sorafenib (NCT03211416), and PDR001 (spartalizumab) plus sorafenib (NCT02988440).

### 5.5. Immune Checkpoint Inhibitors as Neo-Adjuvant or Adjuvant Therapy, or in Combination with Local Treatments

Despite significant improvements in the treatment of early HCC, curative therapies remain associated with high recurrence rates (≈70% at 5 years), and adjuvant therapies able to curb this figure currently represent an unmet need. In both settings of surgery and locoregional treatment, treatment-induced liberation of tumour-associated antigens has previously been demonstrated, thus providing a strong rationale for a combined treatment with immunostimulating agents, as previously shown for other solid tumours [79]. Thus, several studies have been recently initiated in HCC in order to evaluate the safety and efficacy of adjuvant treatments in patients who are at high risk of recurrence after curative hepatic resection or ablation. As an example, a study is currently recruiting patients to test nivolumab against placebo in the adjuvant setting following resection or local ablation (NCT03383458). Similarly, the MK-3475-937/KEYNOTE-937 trial with pembrolizumab is also undergoing in the neoadjuvant setting (NCT03867084). Phase II trials are also evaluating tremelimumab in a similar setting.

Similar to ablation, chemoembolisation has been shown to be associated with enhanced tumour-associated antigens spread together with an increase of vascular endothelial growth factor. In this regard, at least one study on transarterial chemoembolisation plus nivolumab is undergoing (NCT03143270). In this setting, a more complex approach has recently been proposed by the combination of chemoembolisation with both an immune checkpoint inhibitor and a molecular-target agent with an anti-VEGF effect: the LEAP-01 study (combination of chemoembolisation with pembrolizumab and lenvatinib, NCT03713593) and the EMERALD-1 study (combination of chemoembolisation with durvalumab and bevacizumab, NCT03778957). Lastly, transarterial radioembolisation promotes radiation-induced tumour damage similar to that induced by stereotactic radiation therapy: several early studies (Phase I and II) by combining this emergent locoregional approaches to immune checkpoint inhibitors are going to start recruitment (NCT02837029, NCT03033446, NCT03099564, and NCT03380130).

## 6. Liver Involvement in Immune-Related-Adverse Events

Compared to tyrosine-kinase inhibitors as sorafenib and lenvatinib, immunotherapy has significant differences in terms of both toxicity and response. Checkpoint inhibitors are generally better tolerated than tyrosine-kinase inhibitors, although some patients may rarely experience serious, immune-related adverse events involving different organs and systems, such as endocrine glands, the skin, the gastrointestinal tract, the brain and the liver itself [80]. Acute hepatitis is rare, occurring in 4–9% of patients considering all grades of liver injury, and in 3.5% for grade 3 or 4 hepatitis [81,82]. No predictors of checkpoint inhibitors toxicity and immune-related adverse events have been clearly demonstrated. However, the presence of baseline sarcopenia, a family history of autoimmune diseases, tumour infiltration and liver metastases, previous viral infections (such as HIV or hepatitis) and the concomitant use of drugs with autoimmune mechanism of toxicity (anti-arrhythmics, antibiotics, anticonvulsants or antipsychotics) have been suggested to be potential predictors of severe treatment-related toxicity [83,84]. Histological features of the immune-related hepatitis are still little known, due to its rarity and the uncommon utilisation of liver biopsy. A recent French study showed a different histological pattern between patients receiving anti PD-1/PD-L1 and anti-CTLA-4 agents. Anti-CTLA-4-associated injury is typically a granulomatous hepatitis with severe globular necrotic and inflammatory activity and lymphocyte T CD8 cells activation, while the histological pattern of liver damage associated with use of anti-PD-1/PD-L1 agents is more heterogeneous, showing a spotty and confluent necrosis and mild-to-moderate lobular and periportal inflammatory activity, involving both CD4 and CD8 lymphocytes in equal proportions [85]. Finally, three cases of checkpoint inhibitors-induced hepatotoxicity were characterised by biliary injury, and in one patient receiving pembrolizumab for metastatic melanoma a vanishing bile ducts syndrome has been described [86]. According to guidelines, a grade 2 transaminase or bilirubin elevation should prompt the interruption of checkpoint inhibitors therapy and transaminase/bilirubin should be checked twice weekly [80]. A grade 2 elevation lasting for more than two weeks, in the absence of any other cause of liver damage, should be approached with steroids [1 mg/kg/day (methyl)prednisolone or equivalent]. Upon improvement, immunotherapy can be resumed after steroids tapering. Conversely, in the case of worsening, steroids should be increased to 2 mg/kg/day, with permanent discontinuation of checkpoint inhibitors. In the case of grade 3 or 4 transaminase/bilirubin increase, checkpoint inhibitors should be permanently discontinued, steroids must be started (1–2 mg/kg/day) and, if needed, mycophenolate mofetil should be added. In steroid and mycophenolate refractory cases consultation with the hepatologist and liver biopsy are recommended [80].

## 7. Assessment of Treatment Response: The iRECIST Criteria

The response of HCC to immunotherapy appears as low as with tyrosine kinase inhibitors in terms of objective response rates but with longer durability, a finding that appears completely new for this tumour. In particular, the concept of radiological response likely needs a different approach from the one we have used to define treatment response with tyrosine kinase inhibitors: the RECIST and mRECIST criteria will have to be paralleled by a new system specifically designed for these drugs (i.e., the immune-related response criteria, iRECIST) [87].

Indeed, the peculiar tumour response observed with immunotherapy raised questions about the appropriateness of the conventional classification of tumour response, i.e., objective response and disease progression. The RECIST working group has recently developed consensus guidelines for the use of a common language in cancer immunotherapy trials, to ensure consistent design and data collection [87]. The need of a different modality to consider radiological response with checkpoint inhibitors has been raised in early trials in melanoma, when investigators described for the first time a unique response pattern, termed *pseudoprogression*: the disease behaviour met the criteria for disease progression based on RECIST criteria but later patients had marked and durable responses. Thereafter, following a long process of revision of different trials, the major innovation of iRECIST is the concept of resetting the bar if RECIST progression is followed by tumour shrinkage at the subsequent assessment [87]. This evolutive pattern has been defined “unconfirmed progression” (iUPD): if progression is not confirmed, but the tumour shrinks (compared with baseline), which meets the criteria of complete response, partial response or stable disease, then the bar is reset so that iUPD needs to occur again (compared with nadir values) and then be confirmed (by further growth) at the next assessment for confirmed progression (iCPD) to be assigned. Other aspects of lesion assessment are unique to iRECIST. If a new lesion is identified (thus meeting the criteria for iUPD) and the patient is clinically stable, treatment should be continued. Progressive disease is confirmed (iCPD) in the new lesion category if the next imaging assessment, done at 4–8 weeks after iUPD, confirms additional new lesions or a further increase in new lesion size from iUPD (sum of measures increase in new lesion target ≥5 mm, any increase for new lesion non-target).

## 8. Rationale Underlying the Use of the ITA.LI.CA Database to Assess Real-Life Applicability of Checkpoint Inhibitors

With the intent to explore the potential use in clinical practice of two checkpoint inhibitors, nivolumab and pembrolizumab, in HCC patients, we developed some scenarios applying to the Italian Liver Cancer (ITA.LI.CA) database the inclusion criteria adopted by the checkpoint inhibitors studies [48,68]. The ITA.LI.CA database is a large, multi-centre database including patients with newly diagnosed or recurrent HCC approaches managed in a large number of Italian centres with different levels of expertise (secondary and tertiary referral centres) [88]. This database, due to its heterogeneity in terms of tumour stage, underlying liver disease severity, and therapeutic approach, provides a reliable insight into the characteristics of HCC in a Western population, and allows predicting figures of the potential utilisation of these drugs in real-life clinical practice [88,89].

To define the “real world” scenario where these drugs could be used either as up-front or in second-line treatment, we used the selection criteria reported by El-Khoueiry et al. for nivolumab and those proposed by Zhu et al. for pembrolizumab [48,68]. This analysis was mainly aimed at providing the clinicians with a tentative foresight of the proportion of eligible patients and field of applicability of the checkpoint inhibitors in the real-life clinical practice.

### 8.1. First-Line Scenario

To construct the first-line scenarios, we adopted three patient-removal steps: (i) firstly, removal based on the period and type (naive vs. recurrent) of HCC diagnosis; (ii) secondly, removal based on missing data for at least one of the parameters used to identify potential candidates to checkpoint inhibitors; and (iii) lastly, removal based on the selection criteria used for the investigated drug.

As far as nivolumab is concerned, we identified 27 different selection parameters to build the first-line scenario and specifically we considered not amenable to nivolumab patients with one or more of the 27 conditions reported in Table 2, which also reports the number of patients excluded by each step. Thus, we firstly removed patients with recurrent HCC (*n* = 4453) and those with a tumour diagnosis before 2008 (*n* = 3144), and then those in whom data regarding one or more of the 27 selected criteria were missing (*n* = 1403). The remaining 2483 patients with a first diagnosis of HCC over the period 2008–2016 formed the cohort where we tested the amenability to immunotherapy. Among them, 525 patients (21.1%) met the criteria for nivolumab treatment. According to the year of HCC diagnosis, the proportion of potentially treatable cases ranged from 18.3% to 30.3% (Figure 2A), with a median eligibility rate of 20.1% (19.9–20.3% interquartile range).

Considering the eligibility to first-line pembrolizumab, we adopted 30 selection parameters to build the first-line scenario (Table 2). The first two steps were identical to the nivolumab scenario: of the 2483 patients selected by these steps only 268 (10.8%) patients were considered eligible to receive pembrolizumab. Over time, the proportion of patients eligible to pembrolizumab ranged from 9.4% to 21.2% (Figure 2B), with a median eligibility rate of 10.6% (10.2–11.1% interquartile range).

### 8.2. Second-Line Scenario

To build second-line scenarios, we followed the removal steps reported in Table 2. First, we removed the cases diagnoses before 2008 (*n* = 3144) and those with a naive HCC as well as those with ≥2 two recurrences (*n* = 6485). The removal of the 1413 cases with missing data selected 441 patients for nivolumab and 266 patients for pembrolizumab with a first recurrence of HCC after any type of first-line treatment during the period 2008–2016. According to the 27 criteria for nivolumab, only 24 patients (5.4%) resulted eligible for second-line treatment. The proportion of potentially treatable patients ranged from 0% to 10% across the years, with a median of 4.8% (2.9–6.4 interquartile range) (Figure 3A). Likewise, after removing patients with missing data for the 30 variables (*n* = 1588) used for pembrolizumab, only 266 patients with HCC recurrence after any first-line treatment in the period 2008–2016 were selected. Of them, 26 (9.8%) were considered eligible for pembrolizumab treatment, and their proportion ranged over time from 0% to 12.9%, with a median eligibility rate of 8.0% (6.5–10.3 interquartile range) (Figure 3B).

## 9. Conclusions

All the described encouraging results are enriching the scenario of HCC treatment, with a trend to expand the use of immune checkpoint inhibitors, alone or in combination with other molecules, for advanced stage HCCs, as adjuvant therapy after curative approaches in patients with a high risk of disease recurrence, or in combination with transcatheter arterial chemoembolisation in those carrying an intermediate stage HCC.

Nevertheless, despite the expectancy related to ongoing studies, the application of immune checkpoint inhibitors in patients with HCC may still not fulfil the unmet needs of these patients, since as many as 30–40% of them do not respond to these agents, and we have shown by analysing the large ITA.LI.CA database—despite the limitations related to the retrospective nature of the analysis—that in the real-life clinical practice the eligibility rate to immune checkpoint inhibitors is approximately 10–20% in the first-line, and less than 10% in the second-line treatment. The mechanisms of primary resistance to immunotherapy are largely unknown, but combination strategies may overcome this limit, considering that HCC-induced immune tolerance in the setting of a tolerogenic liver environment and chronic inflammation is associated with multiple immunosuppressive mechanisms. Thus, dual or triple combinations of immune targeting agents, associated with inhibitory checkpoint blockage as a backbone of therapy, might be the most promising strategies. Moreover, in this context, it is necessary to identify easily accessible biomarkers to predict tumour response and help us in selecting optimal candidates to immunotherapy. How we will select and monitor these therapies, and use them safely in different groups of patients is not yet clear, as the field is limited by the lack of either tissue or circulating biomarkers to guide clinical decision-making. Additional studies are warranted to identify how many patients (among the whole HCC population, and also among those who undergo this therapy) will actually benefit from immune checkpoint inhibitors treatment, and to assess its cost-effectiveness in this complex disease.

## Figures and Tables

**Figure 1 cancers-11-01689-f001:**
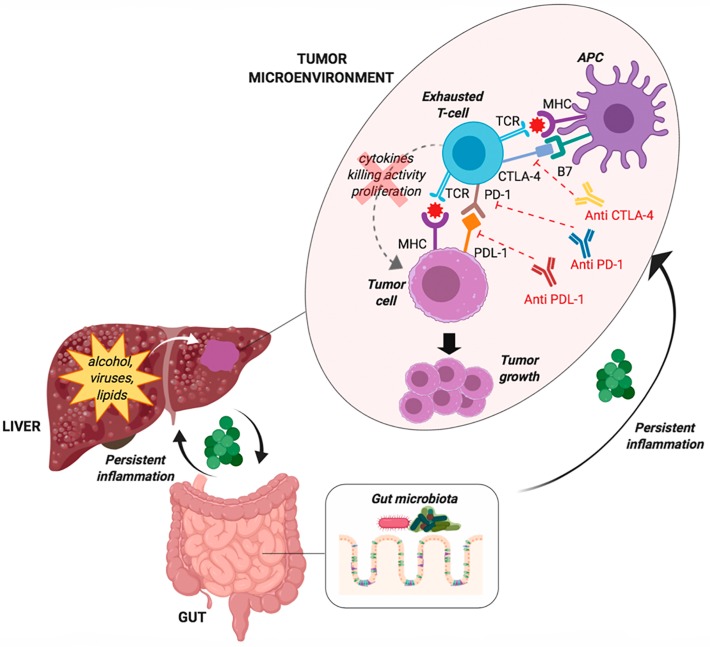
Mechanisms of immune system paralysis in patients with hepatocellular carcinoma (HCC). Inflammatory damage triggered from various factors (alcohol, hepatitis viruses, lipid accumulation, etc.) and from the gut microbiota is involved in the pathogenesis of HCC both directly and indirectly, through T-cells exhaustion. Exhausted T-cells express inhibitory receptor proteins and have a diminished capacity to produce cytokines, proliferate and kill cells. Indeed, antigen presenting cells (APC) and tumour cells express inhibitory molecules such as programmed cell death ligand 1 (PDL-1) and B7 that interact with the surface antigens programmed cell death 1 (PD-1) and cytotoxic T-lymphocyte-associated protein 4 (CTLA-4) on T-lymphocytes, inhibiting the downstream signalling caused by the T-cell receptor (TCR)/ major histocompatibility complex (MHC) interaction with tumour antigens thus favouring tumour growth.

**Figure 2 cancers-11-01689-f002:**
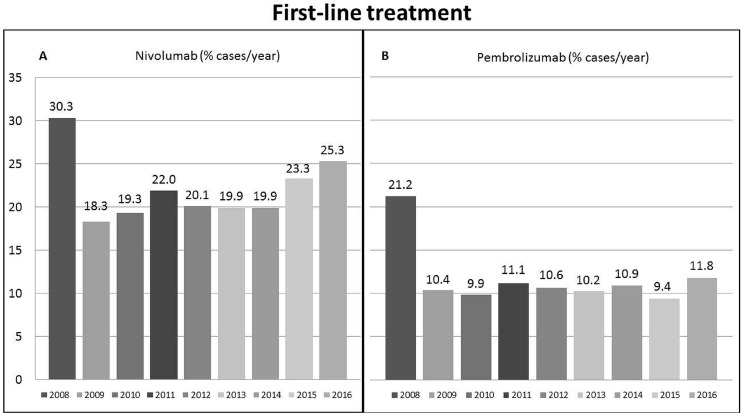
Proportion of patients within the Italian Liver Cancer cohort meeting the criteria for: first-line nivolumab treatment (**A**); and second-line nivolumab treatment (**B**).

**Figure 3 cancers-11-01689-f003:**
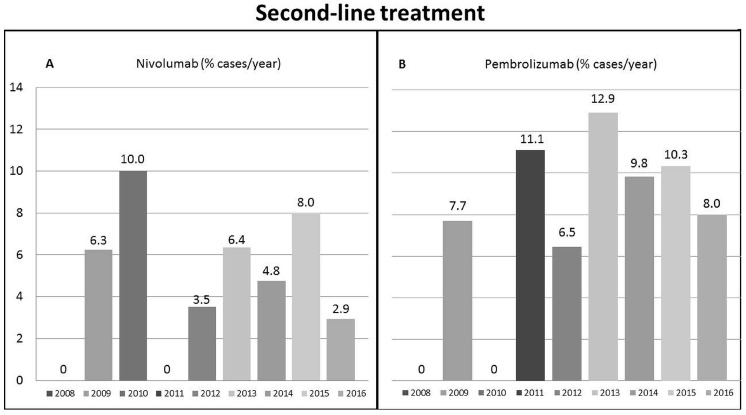
Proportion of patients within the Italian Liver Cancer cohort meeting the criteria for: first-line pembrolizumab treatment (**A**); or second-line pembrolizumab treatment (**B**).

**Table 1 cancers-11-01689-t001:** Ongoing clinical trials exploring immune checkpoint inhibitors: alone, in combination with other drugs or with local therapies.

NCT	Phase	Drug	Procedure	Line of Treatment	Primary End-Point	Estimated Study Completion Date	Company Conducting the Trial
NCT03298451	III	Tremelilumab (+Durvalumab) vs Sorafenib	-	1	OS	06/2021	Astra Zeneca
NCT02576509	III	Nivolumab vs Sorafenib	-	1	OS	07/2020	BMS
NCT03412773	III	Tislelizumab vs Sorafenib	-	1	OS	05/2022	BeiGene
NCT03062358	III	Pembrolizumab vs placebo	-	2	OS	01/2022	MSD
NCT02702401	III	Pembrolizumab vs placebo	-	2	OS, PFS	06/2020	MSD
NCT02702414	II	Pembrolizumab	-	1-2	ORR	05/2021	MSD
NCT02519348	II	Tremelilumab (+Durvalumab)	-	2	Safety, DLT	04/2021	MedImmune LLC
NCT03163992	II	Pembrolizumab	-	2	ORR	12/2020	Samsung Medical Center
NCT02658019	II	Pembrolizumab	-	>2	DCR, Safety	11/2020	Lynn Feun
NCT03389126	II	Avelumab	-	>2	ORR	03/2020	Seoul National University Hospital
NCT03419897	II	Tislelizumab	-	>2	ORR	09/2021	BeiGene
NCT03033446	II	Nivolumab	SIRT	Any	ORR	12/2019	National Cancer Centre, Singapore
ign="middle" style="border-bottom:solid thin">1	ORR	11/2020	Sidney Kimmel Comprehensive Cancer Center at Johns Hopkins
NCT02821754	II	Tremelilumab	Local ablation	1	PFS	04/2021	National Cancer Institute (NCI)
NCT03630640	II	Nivolumab	Electroporation	1	RFS	09/2020	Assistance Publique—Hôpitaux de Paris
NCT03482102	II	Tremelilumab (+Durvalumab)	BRT	2	ORR	10/2025	Massachusetts General Hospital
NCT03316872	II	Pembrolizumab	SBRT	2	ORR	04/2022	University Health Network, Toronto
NCT01658878	IB/II	Nivolumab vs Sorafenib	-	1	ORR	12/2019	BMS
NCT02423343	IB/II	Nivolumab + Galunisertib	-	2	MTD, Safety	12/2019	Eli Lilly and Company
NCT01658878	IB/II	Nivolumab + Ipilimumab	-	>2	ORR	12/2019	BMS
NCT02940496	I/II	Pembrolizumab	-	2	Biomarkers	12/2019	M.D. Anderson Cancer Center
NCT03397654	IB	Pembrolizumab	TACE	1	Safety	12/2020	Imperial College London
NCT02837029	I	Nivolumab	SIRT	Any	MTD	07/2020	Northwestern University
NCT03099564	I	Pembrolizumab	SIRT	1	PFS	01/2020	Autumn McRee, MD
NCT03143270	I	Nivolumab	debTACE	1	Safety	04/2020	Memorial Sloan Kettering Cancer Center
NCT03203304	I	Nivolumab/Ipilimumab	SBRT	1	Safety	08/2020	University of Chicago
NCT01853618	I	Tremelilumab	Local Ablation	1	Safety	12/2020	National Cancer Institute (NCI)

NCT, number of clinical trial (Clinicaltrials.gov); SIRT, selective intra-arterial radiation treatment; MTD, maximum tolerated dose; ORR, overall response rate; PFS, progression free survival; TACE, transarterial chemoembolisation; debTACE, drug eluting beads transarterial chemoembolisation; SBRT, stereotactic body radiation therapy; RFS, recurrence free survival.

**Table 2 cancers-11-01689-t002:** Potential use of nivolumab and pembrolizumab as first-line therapy in HCC patients according tothe ITA.LI.CA database.

**ITA.LI.CA Database**	Number of HCCs = 11,483 (including recurrences)	
(A) First-step removal	1. HCC diagnosis before 01/01/2008 = 3144	
	2. HCC recurrence = 4453	
	Number of patients = 3886 (01/01/2008-31/12/2016)	
(B) Second step removal	Missing data = 1403	
	Examined population = 2483 (100.0%)	
(C) Third step removal	Nivolumab	Pembrolizumab
	Child-Pugh > B7 = 601 ECOG PST > 1 = 343 ECOG PST = 1, BCLC C, resected or RFA/PEI, MC-IN = 86 BCLC 0-A resected = 99 BCLC 0-A RFA/PEI = 238 BCLC B resected = 55 Transplantation = 55 TACE with CR/PR/SD = 577 PBC = 18 Autoimmune hepatitis = 5 Active HBV + HCV = 12 Active HBV + HDV = 12 Autoimmune diseases = 34 Active alcohol abuse = 323 Brain metastases = 2 Story of encephalopathy = 155 Severe ascites = 380 Malignancies previous 3 years = 27 HIV = 22 Leucocytes < 2000/mcL = 63 PLT < 60,000/mcL = 299 Hb < 9 g/dL = 107 GFR < 40 mL/min = 147 Total bilirubin > 3.0 mg/dL = 214 AST/ALT > 5× = 123 Albumin < 2.8 g/dL = 226 INR > 2.3 = 34	Child-Pugh > B7 = 601 ECOG PST > 1 = 343 ECOG PST = 1, BCLC C, resected or RFA/PEI, MC-IN = 86 BCLC 0-A resected = 99 BCLC 0-A RFA/PEI = 238 BCLC B resected = 55 Transplantation = 55 TACE with CR/PR/SD = 577 PBC = 18 Autoimmune hepatitis = 5 Active HBV = 95 Double infection HBV/HCV = 36 Autoimmune diseases = 34 Active alcohol abuse = 323 Brain metastases = 2 Story of encephalopathy = 155 Clinically apparent ascites = 1009 Malignancies previous 5 years = 43 HIV = 22 Leucocytes < 1200/mcL = 23 PLT < 60,000/mcL = 299 Hb < 8 g/dL = 33 sCr > 1.5 mg/dL = 121 GFR < 60 mL/min if sCr < 1.5 mg/dL = 502 Total bilirubin > 2.0 mg/dL = 440 AST/ALT > 5× = 123 Albumin < 3.0 mg/dL = 414 INR > 1.5× = 60 Variceal bleeding < 6 months = 103 Main branch PVT/IVC thrombosis = 187
	Final population = 525/2483 (21.1%)	Final population = 268/2483 (10.8%)

Abbreviations: ITA.LI.CA, Italian Liver Cancer; HCC, hepatocellular cancer; ECOG, Eastern Cooperative Oncology Group; PST, performance status; BCLC, Barcelona Clinic Liver Cancer; RFA, radio-frequency ablation; PEI, percutaneous ethanol injection; MC, Milan Criteria; TACE, trans-arterial chemoembolisation; CR, complete response; PR, partial response; SD, stable disease; PBC, primitive biliary cholangitis; HBV, hepatitis B virus; HCV, hepatitis C virus; HDV, hepatitis D virus; HIV, human immunodeficiency virus; PLT, platelets; Hb, hemoglobin; GFR, glomerular filtration rate; sCr, serum creatinine; AST, aspartate transaminases; ALT, alanine transaminases; INR, international normalised ratio; PVT, portal vein thrombosis; IVC, inferior vena cava.

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
