# Peer review of "Overview of Immune Checkpoint Inhibitors Therapy for Hepatocellular Carcinoma, and The ITA.LI.CA Cohort Derived Estimate of Amenability Rate to Immune Checkpoint Inhibitors in Clinical Practice"

_cancers, 2019, doi:10.3390/cancers11111689_

Round 1

Reviewer 1 Report

the authors satisfactorily responded to my previous comments

Reviewer 2 Report

I reviewed the manuscript and found that the author has seriously considered our critics and revised the manuscript as per our suggestions for example, regarding 1st comment, about the ambiguity in the main and the content inside the text, the author has modified the main title. Regarding 3rd comment, the author has revised the figure-1 and addressed the missing “Exhausted T-cell” point in line #249. Regarding 4th comment, the author has also revised Table-1 and addressed the missing company-name and appropriate (*) highlighting in the line #327. All other comments were also revised as advised. Overall, now I feel this manuscript is satisfactory for publication in the Cancers.

This manuscript is a resubmission of an earlier submission. The following is a list of the peer review reports and author responses from that submission.

Round 1

Reviewer 1 Report

A well written manuscript on IO therapy in HCC, correctly summarizing the current data base.

It is separated into two segments:
1. analysis of the existing and ongoing clinical trials using IOs in HCC - this is appropriate and I don't have comments.

2. An attempt to use the ITA.LI.CA database to assess real life applicability of Nivolumab or Pembrolizumab. Basically, the authors take out patients from this registry because of a lack of data or because they don't fulfill the inclusion criteria of the respective trials.

This ssems to me arbitrary and no basis to make clinical decisions

Author Response

Please, see attachment

Reviewer 2 Report

The first two paragraphs in the Introduction are two broad.

Best to discuss hepatoma, rising incidence etc, rather than the rising incidence of cancer in general.

Could benefit from  greater discussion on how to predict response to checkpoint inhibitors in HCC.

e.g  PD L-1 staining? baseline TGF-beta levels (Cancer 2019 Jun 28)? etc.

Reviewer 3 Report

Manuscript Number: cancers-583478-peer-review-v1

Title: Overview of immunotherapy for hepatocellular carcinoma, and the ITA.LI.CA cohort derived estimate of amenability rate to immunotherapy in clinical practices

The review written by Giannini et al., on “Overview of immunotherapy for hepatocellular carcinoma, and the ITA.LI.CA cohort derived estimate of amenability rate to immunotherapy in clinical practices” seems somehow satisfactory and informative, but following points should be reconsidered for a better visibility to readers:

Major points

The authors have tried to justify the immunotherapeutic approaches for HCC, and analyzed ITA.LI.CA cohort derived from the database of their country (Italy). For that, authors have described into several subsections. But considering the clarity and the main goal of this review, I feel the main title, abstract and the main content are not co-relating to each other. For example, in the main title they have written “Overview of immunotherapy for hepatocellular carcinoma” but inside the content they have described mostly about immune checkpoint blockade antibody therapy that is only a part of immunotherapy. So the title should be revised appropriately. Cell-based immunotherapy, such as dendritic cell vaccine or NK cell therapy, for HCC should be described in Section 2 or 3 or both with appropriate references. In figure-1, the “figure legend” is justifying the mechanism of immune paralysis in HCC patients by alcohol, viruses and lipid accumulation. The author should reconsider to revising the picture, for example T-cell exhaustion is not described in the picture for the main reason of immune paralysis in HCC in the text. In table-1, the clinical trials which were already completed in February, April, June and July of 2019, it would be more interesting if the authors would mention the place and name of the company conducting this trial (if available). In the same table, why the authors have highlighted (*) the placebo at one place only! See page #8, 4th and 6th line of 3rd column. In section 5.2. ‘Results of monotherapy with checkpoint inhibitors in HCC’, the authors want to show the result of monotherapy. Many readers wish to look for the final response of the immunotherapeutics together with satirical score. Authors are asked to stick to their conclusion for a better understanding. In addition, authors have mentioned six-antibodies in monotherapy for HCC, but the descriptions for each monotherapy are not consistent, such as; somewhere they have mentioned antibody dose and somewhere % response to/and number of total patients taken for the trial, please be consistent to each therapy to keep the readers’ attention.  I am not much convinced with the sub-section 5.3. ‘Combination of two immune checkpoint inhibitors’ has only fascinating title. Authors should consider the comments above for this section too. I could not find any clear take-home message or attractive conclusion from these combination therapies especially with Nivolumab plus Ipilimumab, rather than deviating from the goal! Same suggestions (given above in comment 5-6) for sub-section 5.4. ‘Combination of immune checkpoint inhibitors with molecular-targeted agents’ also. I would suggest authors to revise this important section. Section 5.5 should be expanded with appropriate references. At least in one section among sections 5.3 to 5.5, one or two paragraphs should be added highlighting the outcome of newly introduced combination therapies being used in clinical trials nowadays for readers’ attraction. I think it would be better for Section 5.6 and 5.7 to be written in a separate Section 6 and 7, respectively. 11. I would suggest the authors to recompose the Section 5.8 to 5.10 in a separate Section, such as Section 8 and subsections for the ITA.LI.CA database and selection criteria for first- and second-line treatments. In the beginning of the section, the purpose and expected outcomes of this study would be appreciated by readers and other researchers. ITA.LI.CA should be included in the conclusion.

Minot points

Abstract needs be fully revised for grammatical mistakes. line #61 among the keywords “check-point inhibitors; cirrhosis; immunology; outcome” ‘HCC’ instead of ‘cirrhosis’; immunology seems not appropriate line #67, “9.6 million deaths in 2018.1 is cited by reference 1, but the reference is not clear. line #70, ‘histopathological’ would be more appropriate than ‘histomorphological’ Line # 96: entrains should be revised line #101, “…agents are used to boost…” is also not clear. Please, describe the agents or at least brief category of the agents (drugs) in this sentence. lines #101-102, what are those natural mechanisms? natural body defense mechanisms? lines #140-144, clarify the cited reference for each sentence, because their references (14-17) justify the first-line and second-line treatment not for the ‘somehow overlapping, molecular mechanisms of action’. The suitable references for the ‘somehow overlapping MOA’ should be included separately. lines #161-163, “PD-1 is –--- and other cell types--- and proliferation” this sentence needs specific reference(s). Specify the cell types instead of ‘other cell types’ line #281, ‘reseach’ spelling error. Line # 509-513: one paragraph is just one sentence! Needs to be trimmed down!
